# Management of Unresectable Localized Pelvic Bone Sarcomas: Current Practice and Future Perspectives

**DOI:** 10.3390/cancers14102546

**Published:** 2022-05-22

**Authors:** Joaquim Soares do Brito, Miguel Esperança-Martins, André Abrunhosa-Branquinho, Cecilia Melo-Alvim, Raquel Lopes-Brás, João Janeiro, Dolores Lopez-Presa, Isabel Fernandes, José Portela, Luis Costa

**Affiliations:** 1Department of Orthopedics, Centro Hospitalar Universitário Lisboa Norte, 1649-028 Lisboa, Portugal; jotaportela@gmail.com; 2Department of Medical Oncology, Centro Hospitalar Universitário Lisboa Norte, 1649-028 Lisboa, Portugal; cecilia.alvim.moreira@gmail.com (C.M.-A.); raquel.bras@chln.min-saude.pt (R.L.-B.); 15673@chln.min-saude.pt (I.F.); luis.costa@chln.min-saude.pt (L.C.); 3Luís Costa Lab, Instituto de Medicina Molecular–João Lobo Antunes, Faculdade de Medicina de Lisboa, 1649-028 Lisboa, Portugal; 4Department of Radiotherapy, Centro Hospitalar Universitário Lisboa Norte, 1649-028 Lisboa, Portugal; andre.branquinho@chln.min-saude.pt; 5Department of Radiology, Centro Hospitalar Universitário Lisboa Norte, 1649-028 Lisboa, Portugal; joao_janeiro@sapo.pt; 6Department of Pathology, Centro Hospitalar Universitário Lisboa Norte, 1649-028 Lisboa, Portugal; doloreslopez@chln.min-saude.pt

**Keywords:** management, pelvic bone sarcoma, treatment option, unresectable bone sarcoma

## Abstract

**Simple Summary:**

Some locally advanced pelvic bone tumors are deemed unresectable and, as such, not suitable for curative surgery. In this setting, treatment options are generally limited and not unanimous, with decisions being made on an individual basis after multidisciplinary discussion. Ultimately, and notwithstanding the bright prospects raised by novel therapeutic approaches, treatment should be patient-tailored, weighing a panoply of patient- and tumor-related factors.

**Abstract:**

Bone sarcomas (BS) are rare mesenchymal tumors usually located in the extremities and pelvis. While surgical resection is the cornerstone of curative treatment, some locally advanced tumors are deemed unresectable and hence not suitable for curative intent. This is often true for pelvic sarcoma due to anatomic complexity and proximity to vital structures, making treatment options for these tumors generally limited and not unanimous, with decisions being made on an individual basis after multidisciplinary discussion. Several studies have been published in recent years focusing on innovative treatment options for patients with locally advanced sarcoma not amenable to local surgery. The present article reviews the evidence regarding the treatment of patients with locally advanced and unresectable pelvic BS, with the goal of providing an overview of treatment options for the main BS histologic subtypes involving this anatomic area and exploring future therapeutic perspectives. The management of unresectable localized pelvic BS represents a major challenge and is hampered by the lack of comprehensive and standardized guidelines. As such, the optimal treatment needs to be individually tailored, weighing a panoply of patient- and tumor-related factors. Despite the bright prospects raised by novel therapeutic approaches, the role of each treatment option in the therapeutic armamentarium of these patients requires solid clinical evidence before becoming fully established.

## 1. Introduction

Bone sarcomas (BSs) are rare mesenchymal neoplasms with an estimated incidence of 0.8 per 100,000 patients per year in Europe [1]. Among these, pelvic bone sarcomas (PBSs) represent a particular management challenge, as they are often large in size and typically enclosed by important organs, major blood vessels, and nerves [2,3,4,5]. It is well-documented that these tumors have poorer prognoses compared to those arising in other locations [2,3,4]. Nonetheless, complete surgical resection is the cornerstone of integrated multimodal curative treatment [6], as shown by the overall survival (OS) rates reported in surgically treated PBSs patients, far superior to those of patients not submitted to surgical resection [6].

Locally advanced PBSs may be particularly bulky and infiltrative and involve important anatomic structures, and hence they may not be amenable to curative surgery. As such, treatment options for PBSs deemed unresectable are generally limited [5].

In recent years, several studies have been published exploring innovative treatment options for patients with locally advanced sarcomas not amenable to local surgery [6,7,8,9,10]. These tumors, in general, and PBSs in particular, comprise a heterogeneous group requiring specific and individualized treatment approaches. This article reviews the state-of-the-art regarding management strategies for unresectable PBSs and explores in further detail specific treatment modalities for each main histologic subtype.

The information regarding specific treatment modalities for each PBS histologic subtype is summarized and organized in tables, which may be found at the end of each respective section. To more systematically present current evidence on the clinical utility of every management method, the authors decided to assign a level of relevance to each therapeutic intervention (these levels of relevance are displayed in the mentioned tables). As such, the treatment relevance levels (*not relevant, scarcely relevant, relevant*, and *highly relevant*) were attributed by the authors according to particular criteria: interventions only tested in pre-clinical studies were labeled as *not relevant* or *scarcely relevant*, keeping in mind the current lack of application of those interventions in clinical practice, with the two possible levels distinguishing the degree of relevance of interventions evaluated in pre-clinical studies as an attempt to reflect how promising these interventions seem to be based on the results from those studies. Interventions tested in clinical studies were labeled as *not relevant* if no benefits in overall survival (OS) or progression-free survival (PFS) were observed. Interventions tested in clinical studies were differentially labeled as *scarcely relevant*, *relevant* or *highly relevant* on the basis of the phase levels of the studies where potential benefits fop, those interventions were shown and the degree of OS and/or PFS benefit produced by the use of those interventions. Nevertheless, the authors underline that this is a subjective score based on conclusions reported by the authors of the different studies herein presented. This classification attempts to provide valid and clinically oriented information for practical application.

## 2. Definition of Unresectable Pelvic Sarcoma

The pelvic anatomy poses great challenges to orthopedic surgeons [11]. A high level of anatomic knowledge is required when addressing BSs arising at this topography, given the presence of a significant number of important organs, main blood vessels, and crucial nerve structures packed in a tight space [12]. In addition to the hazard associated with the contiguity of vital anatomic structures, the complexity of the pelvic anatomy usually requires intricate reconstruction [12]. 

As mentioned above, PBSs often present with significant size and remarkable extension to neighboring organs, vessels, and nerves. This fact, together with the convoluted anatomy of the pelvis, makes clear-margin resection particularly difficult to achieve [13]. In general terms, we can consider as surgically unresectable any PBS where a clear negative margin cannot be achieved, where the surgical morbidity will be too extreme to accept comparison with potential advantages, or where the patient does not accept the functional impairment and expected complications generated by the surgical procedure (Figure 1 and Figure 2) [14]. Given this, alternative therapeutic options cannot be overlooked and should be discussed within a multidisciplinary sarcoma team meeting. 

## 3. Radiotherapy and Its Different Modalities: A Cornerstone of Unresectable Pelvic Sarcoma Management

Radiation therapy is widely used for the treatment of different unresectable pelvic sarcomas, being a transversal treatment option for the most common histologic subtypes of pelvic sarcomas. The preponderance of its role in their management strategy is growing with the use of several different modalities. 

External beam radiotherapy (EBRT) is an option when treating unresectable cancer and relies on delivering a specific ionizing radiation (photons or particles) to a target volume (i.e., a tumor and its subclinical affected area) while avoiding a potential harmful dose onto the surrounding healthy organs at risk (OARs). Achieving this goal depends on multiple factors (e.g., clinical indication, personnel expertise, treatment planning system and linear accelerator capabilities and accessibility). EBRT technologies can subdivided in terms of the quality of the radiation beam (photons or particles) and the technology delivery approach (Figure 3) [15].

In the current state, conventional linear accelerators (available in any radiotherapy department) can produce photons and electrons with various technical-planning delivery approaches for photons: 3D planning radiation therapy (3D-CRT), intensity-modulated radiotherapy (IMRT), volumetric modulated arc therapy (VMAT), and stereotactic radiotherapy approaches. Moreover, hadrontherapy, also known as heavy particle therapy (protons or carbon ions), is available in a few specific facilities where heavy particles can be accelerated through cyclotrons or synchrotrons. Beam arrangement, modulation, and delivery for hadrontherapy are still evolving, especially for protons (e.g., “pencil beam scanning” for intensity-modulated proton therapy). Table 1 explains some definitions [16].

The rationale of choosing hadrontherapy over relying on photon beam technology when treating unresectable or midline pelvic bone is based on physical and radiobiologic properties [16,17]:The Bragg peak effect in which the dose distribution can be fully released in depth inside the target while a limited amount is deposited when entering tissue and after the sharp dose release in the tissue. This an added value when treating patients with midline tumors or with nearby OARs, because photons cannot be modulated to confer appropriate protection when dose escalation is needed.The higher linear energy transfer (LET) than photons, which leads to a higher relative biological effectiveness (RBE), where DNA damage is higher and might not be dependent on hypoxia tumor status. This is useful in tumors that are conventionally “radioresistant”, namely certain soft tissue sarcoma and bone cancers.

## 4. Osteosarcoma

Osteosarcoma is the most common primary malignant tumor of bone [18]. Most cases occur in children and young adults, with a second, smaller incidence peak in patients over 60 years, often associated with Paget’s disease [19]. Osteosarcoma typically affects long bones but can also present in the pelvic region, most often involving the proximal femur (5%) and iliac bone (3%) and very rarely the sacrum, pubic, and ischial bones [20]. In the pelvic region, it represents the second most common primary bone tumor, along with Ewing sarcoma and after chondrosarcoma, accounting for approximately 20% of primary pelvic bone malignancies [21,22]. Although most osteosarcoma patients (>85%) have localized disease at diagnosis, pelvic osteosarcoma often presents late in the course of the disease, with already large tumors and metastatic spread (mostly to the lungs) [3,19]. 

Conventional osteosarcoma is the most frequent form of osteosarcoma, identified in 70–80% of cases, with the chondroblastic variant being more frequently found in the pelvis than in the appendicular skeleton and showing intrinsic higher resistance to chemotherapy (ChT) [5]. Osteosarcoma diagnosis is based on the identification of neoplastic bone cells. Neoplastic cells may be fusiform, epithelioid, or plasmacytoid, typically showing severe anaplasia. The presence of woven bone or osteoid (produced by malignant cells) in close association with surrounding malignant cells is required for diagnosis. Variable amounts of chondroblastic and fibroblastic components can be also be found, with osteosarcomas being classified as osteoblastic, chondroblastic, or fibroblastic based on the predominant component [5]. If the chondroblastic component is predominant, as observed in chondroblastic and periosteal osteosarcoma, the differential diagnosis with chondrosarcoma can be challenging, especially in core biopsy analysis. The immunohistochemistry profile lacks specificity and is commonly characterized by SATB2 [20], osteonectin, osteoprotegerin, and osteopontin expression, but keratins and epithelial membrane antigen (EMA) may also be expressed, potentially representing a diagnostic pitfall. 

The most common symptom associated with pelvic osteosarcoma is recurrent bone pain, which frequently prompts imaging studies that typically show radiologic features similar to those found in osteosarcoma of the extremities (osteoid formation, asymmetric bone destruction, and asymmetric soft tissue extension) [19,20]. The cartilaginous component (either in conventional or chondroblastic variants) is identified by ring and arc enhancement in gadolinium contrast-enhanced magnetic resonance imaging (MRI) [20]. Imagiologically, the most important differential diagnoses include chondrosarcoma and Ewing sarcoma, as these bone tumors also display large cartilaginous components.

Osteosarcoma has some of the poorest outcomes and worst prognoses among primary pelvic bone tumors, with 5 year survival rates of 19% compared with 45% for pelvic sarcoma in general [5]. This is the result of typically late symptomatic presentation, frequent metastatic disease at diagnosis, intrinsic surgical challenges secondary to the complex pelvic anatomy, and characteristic chemoresistance of chondroblastic variants arising at this location [5]. Other factors affecting prognosis include tumor grade and size, type of surgery, surgical margins, and patients’ age [21]. An optimal surgical approach by a trained surgical team and ChT are the mainstays of treatment. 

ChT consists of combination regimens administered before and after surgery, with high-dose methotrexate, doxorubicin, and cisplatin (MAP) being the preferred regimen for adolescents and young adults [23,24,25]. High-dose methotrexate is usually waived in older patients due to high toxicity. Alternatives to doxorubicin and cisplatin include ifosfamide and etoposide, but neither has proven superior to MAP [19]. 

Patients with unresectable localized disease or metastases at diagnosis should be assessed by a multidisciplinary team for potential resectability following systemic treatment, based on imagiological reevaluation with computed tomography (CT) and MRI after neoadjuvant ChT (Table 2) [24]. 

**Table 2 cancers-14-02546-t002:** Summary of therapeutic options for the management of unresectable pelvic osteosarcoma: Evidence from clinical studies.

Therapeutic Modality	Therapeutic Relevance	Evidence Level	Comments
Particle radiotherapy (carbon ions; protons/photons) [26,27,28,29]	+++	II	Data obtained from studies not specifically designed for pelvic osteosarcoma
Gemcitabine + docetaxel [30]	++	IV	
Multi-drug chemotherapy + radiotherapy [31]	+++	IV	
Sorafenib [32]	+++	III	
Regorafenib [33]	+++	II	
Cabozantinib [34]	+++	II	
Apatinib [35,36]	+++	III/IV	
Pazopanib [37]	+++	IV	
Sorafenib + everolimus [32]	+++	III	
Robatumumab [38]	+	II	
Pembrolizumab + cyclofosfamide [39]	+	III	
Embolization [40]	+++	IV	Relevant for pain control
(153)Sm-EDTMP [41]	+++	IV	Relevant for pain control

+ Not relevant; ++ scarcely relevant; +++ relevant. (153)Sm-EDTMP, samarium-153 ethylene diamine tetramethylene phosphonate.

Radiotherapy (RT) should be offered to patients with tumors deemed unresectable at primary setting or after neoadjuvant treatment, allowing a posterior surgical approach in some of these patients [23,24,26,27]. RT can also be offered after incomplete resection to improve local control [26,27]. Particle therapy with carbons, protons, or protons and photons seems to achieve promising local control rates, but the available data have not been specifically obtained for pelvic osteosarcoma, and further studies are required to support the efficacy of this approach in these tumors and define the optimal setting for its use [26,27]. 

In metastatic settings, the first-line ChT regimen is similar to the one used in neoadjuvant settings, while second-line ChT includes combinations of agents with known activity against osteosarcoma, like gemcitabine, docetaxel, cyclophosphamide, carboplatin, and topotecan. The combination of gemcitabine and docetaxel is frequently considered an option for unresectable and metastatic osteosarcoma. Palmerini et al. [24] reported a median progression-free survival (PFS) of 3.5 months and a 4 month PFS of 46% with this combination. On the other hand, Hernberg et al. [24] showed that multidrug ChT regimens used in combination with RT can also be effective for symptomatic control and provide appreciable local control rates in selected patients. 

The multikinase inhibitor sorafenib was the first tyrosine kinase inhibitor (TKI) to show activity in osteosarcoma [32,42]. Recent studies with regorafenib [33], cabozantinib [34], apatinib, and pazopanib confirmed that these agents can also play a role in osteosarcoma management, particularly in unresectable or metastatic settings, by delaying disease progression, although they do not significantly impact OS [32,42]. Grignani et al. explored sorafenib and the combination of sorafenib with the mammalian target of rapamycin (mTOR) inhibitor everolimus [32,42] in patients with unresectable high-grade osteosarcoma and reported promising PFS rates at five months. A phase II study by Anderson et al. evaluated the role of targeted therapies in this setting, reporting disappointing results with robatumumab, an insulin-like growth factor receptor 1 (IGF-1R) inhibitor [34]. Immunotherapy was also investigated in unresectable osteosarcoma, with poor outcomes even when combined with standard ChT [34].

Tumor embolization represents another possible option for unresectable pelvic osteosarcoma, having shown effectiveness in local control and pain management in small patient series [34]. The beta-emitting bone-targeted samarium-153 ethylene diamine tetramethylene phosphonate ((153)Sm-EDTMP) was evaluated as a pain control agent and shown to have a potential role in the management of bone pain in patients with osteoblastic metastases [41]. However, its use has not been widely adopted.

## 5. Chondrosarcoma

Chondrosarcoma (CSs) are a heterogeneous group of cartilage-producing tumors and one of the most common primary malignancies of bone, accounting for 20–27% of all new primary malignant osseous neoplasms [43,44]. A discrete male predominance and median age at diagnosis of 50 years old have been reported [6,43,44]. These neoplasms can be classified according to the location where they arise, with the pelvis and proximal femur being the most common primary origin sites, and according to the presence of precursor lesions, being classified as primary when arising *de novo* and as secondary when arising from preexisting lesions, mostly osteochondromas or enchondromas [6,44]. The clinical behavior is variable and predicted by the histologic grade, with most CSs being low-grade tumors and thus presenting a very slow-growing pattern and favorable prognosis. Still, a small proportion of CSs are high-grade tumors and carry a significant risk of development of metastases, with associated dismal prognosis [6,43,44,45]. Histologically, CS is a lobulated neoplasm with a hyaline or myxoid cartilaginous matrix. Cellularity and cytological atypia increase with grade, as well as the number of mitoses, and cells are usually positive for S100 [46]. For grade 1 CS, the differential diagnosis should always comprise enchondroma.

Regarding treatment, the only option with curative intent is complete surgical resection. However, in specific locations, such as the pelvis or skull, wide margin resection is difficult to achieve and negatively impacts disease prognosis, underlining the importance of developing new local and systemic approaches to improve the outcomes of patients with advanced or unresectable CS (Table 3 and Table 4) [45,47].

**Table 3 cancers-14-02546-t003:** Summary of therapeutic options for the management of unresectable pelvic chondrosarcoma: evidence from clinical studies.

Therapeutic Modality	Therapeutic Relevance	Evidence Level	Comments
Photon bean radiotherapy [48]	+++	IV/III	
Proton bean radiotherapy [48,49]	+++	IV/III	
Carbon ions radiotherapy [50,51]	++++	III/IV/IV	
Chemotherapy [52,53,54]	++	IV/IV/IV	With particular interest for mesenchymal and dedifferentiated chondrosarcoma
Pazopanib [9]	+++	III	
Regorafenib [55]	++	II	
Ramucirumab [56]	+++	IV	
Ivosidenib [57]	+++	III	Option for IDH1-mutant chondrosarcomas
Palbociclib [58,59]	+++	III/IV	
Sirolimus + cyclophosphamide [60]	+++	IV	Lymphopenia observed in 50% of patients
Pembrolizumab [61]	++	III	Only five patients with chondrosarcoma in the study

++ scarcely relevant; +++ relevant; ++++ highly relevant. IDH1, isocitrate dehydrogenase 1.

**Table 4 cancers-14-02546-t004:** Summary of therapeutic options for the management of unresectable pelvic chondrosarcoma: evidence from pre-clinical studies.

Therapeutic Modality	Therapeutic Relevance	Evidence Level	Comments
Everolimus [62]	++	II	Non-synergistic association with doxorubicin

++ scarcely relevant.

RT is an option for CSs ineligible for surgical resection, providing local control and symptomatic relief, especially for mesenchymal CS due to its increased radiosensitivity [47,63]. The current evidence has been mainly retrieved from national databases, unicentric retrospective studies, and prospective phase I-II trials. The analysis of the 2019 American National Cancer Database showed that a ≥70 Gy dose of definitive radiotherapy (DRT) conferred improved 5 year OS (86.3% vs. 69.2%; *p* = 0.009), although only 28 of 175 patients have received the treatment [48]. Besides photonic techniques, the use of proton bean irradiation is also being evaluated for its potential benefits, such as allowing the delivery of high doses of radiation in the target volume while limiting the dose in surrounding tissues, leading to less severe late side effects by exploiting the Bragg peak effect [49,64]. In the same analysis, proton therapy was shown to confer higher 5 year OS (75.0% vs. 19.1%; *p* = 0.007). However, in subgroup analyses, the use of proton therapy lost significance for a 5 year OS benefit (75.0% vs. 33.1%; *p* = 0.090). It can be hypothesized that these results were due to the low number of patients treated with protons in the DRT setting in this study (6 in 175) [48]. The use of carbon ions (another type of hadrontherapy) represents another treatment option, which combines the physical advantages of heavy particles (Bragg peak effect) with high linear energy transfer, resulting in peculiar radiobiological activity, increased cell death activity in hypoxic tumors, and a relative drop of biological effectiveness against tumors at high doses/fractions [50,65]. It should be noted that most data regarding the treatment of pelvic tumors with carbon ions do not distinguish between different tumor histologies (also including chordomas, among others), precluding robust conclusions for these tumors. Still, Outani et al. published a retrospective analysis comparing overall local recurrence and OS in 31 patients with pelvic CS submitted to surgery (*n* = 24) or treatment with carbon ions at 70.4 GyE in 16 fractions (*n* = 7) [51]. Neither OS (*p* = 0.347) nor local recurrence (*p* = 0.932) significantly differed between treatment groups, but patients who underwent surgery had impaired function compared with those who were irradiated (*p* = 0.03). Notwithstanding the potential biases, data suggest that RT can be a local treatment option for inoperable cases and should be discussed in multidisciplinary board meetings to determine the most appropriate RT alternative. Imai et al. reported the effectiveness of carbon ion RT in patients with unresectable CS, also showing that local control rates were dependent on grading and histologic subtype [50]. Demizu et al. reported encouraging effectiveness of particle therapy with protons or carbon ions in unresectable pelvic sarcoma, despite the low number of individual sarcoma subtypes included in the study and short follow-up [66].

ChT has been considered ineffective in CS, especially in low-grade tumors, due to their slow-growing pattern and low fraction of cell division [6,44]. Other possible reasons for CS chemoresistance are the high activity of anti-apoptotic and pro-survival pathways, with expression of Bcl-2 family proteins, and reduced intracellular access of ChT due to expression of multidrug resistance 1 gene (P-glycoprotein), poor vascularity, and abundant extracellular matrix [44]. Given this chemoresistance, enrollment of CS patients in clinical trials is mandatory, especially in the advanced/metastatic setting. Nonetheless, patients with mesenchymal and dedifferentiated CS seem to benefit more from ChT than those with conventional CS [52]. In the absence of clinical trials, the chemotherapeutic protocols used in advanced/metastatic mesenchymal CS are extrapolated from Ewing sarcoma and consist of vincristine, doxorubicin, and cyclophosphamide (VDC) alternating with ifosfamide and etoposide (IE), while the protocols for advanced/metastatic dedifferentiated CS rely on data extrapolated from osteosarcoma and comprise cisplatin and doxorubicin. To date, the biggest benefit observed was a PFS gain of 2.5 months in high-grade CS patients treated with doxorubicin monotherapy and of 3.6 months in patients treated with the combination of doxorubicin and cisplatin. No OS benefit has been reported yet [44,53,54].

Other options are being explored for patients refractory to ChT, including angiogenesis inhibitors due to evidence of pathologic neovascularization in cartilaginous tumors. The TKI pazopanib demonstrated efficacy in phase II trials of patients with unresectable or metastatic CS, with a disease control rate at 16 weeks of 43% and median OS and PFS of 18 and 8 months, respectively [9,44]. Ramucirumab, a monoclonal antibody targeting vascular endothelial grow factor receptor 2 (VEGFR2), showed partial long-lasting disease stabilization (over 6 months) in metastatic CS in small trials [56]. Despite having failed its primary endpoint of PFS at 12 weeks, a phase II trial of regorafenib suggested that this oral multikinase inhibitor targeting tumor angiogenesis and the microenvironment may slow disease progression in patients with metastatic CS after failure to prior ChT [55].

Other molecular targeted therapies are under assessment in CS. Around 40–56% of CS have been reported to have mutations in isocitrate dehydrogenase (IDH) enzymes as an early event in disease development, causing the accumulation of the D-2-hydroxyglutarate (2-HG) oncometabolite [44,57], which prompted research on IDH inhibitors. A phase I trial of ivosidenib, a selective inhibitor of mutant IDH1 enzyme, showed minimal toxicity, substantial 2-HG reduction, and durable disease control in patients with advanced mutant-IDH1 CS [57].

Cyclin-dependent kinase (CDK) inhibitors are a class of molecularly targeted therapies widely used in the treatment of advanced breast cancer that have shown activity in lipomatous soft tissue sarcoma. A clinical study evaluated CDK4 expression in tissue samples of CS patients and showed that expression levels were associated with the development of metastases and disease recurrence. Treatment with palbociclib led to CDK4 attenuation, inhibiting CS cell viability via CDK 4/retinoblastoma (Rb) signaling pathway regulation and highlighting the promising use of CDK4 inhibitors in CS treatment [44,58,59]. A phase II trial of abemaciclib is being conducted in patients with confirmed diagnosis of metastatic or unresectable soft tissue sarcoma or BS including CS [44].

The mTOR pathway is commonly activated in CS cell lines, with phosphorylation of S6, a downstream marker of mTOR activity, being activated in up to 69% of conventional and 44% of dedifferentiated CS [44]. A study investigating the combined use of everolimus and doxorubicin showed no synergistic effect, but everolimus alone did have a suppressive effect on the tumor [62]. Another observational study of 10 patients with unresectable CS treated with sirolimus in combination with cyclophosphamide resulted in a median PFS of 13.4 months, with an objective response seen in one patient and stable disease for at least six months in six patients. Despite these results, the side effects associated with this treatment are a concern, with grade 3–4 adverse events (mostly lymphopenia) reported in almost 50% of patients [60].

As in other tumor types, immunotherapy represents a potential new therapeutic approach in CS, with programmed cell death ligand-1 (PD-L1) overexpression observed in approximately 41% of dedifferentiated tumors [44]. In the phase II SARC028 trial, 86 patients with unresectable or metastatic sarcoma of various histologies, five of whom with dedifferentiated CS, were treated with the anti-programmed cell death protein-1 (PD-1) pembrolizumab [61]. Twenty percent of these patients (i.e., one in five) achieved an objective response. Several trials are currently ongoing exploring the use of immune checkpoints inhibitors (ICIs) in monotherapy or in combination with ChT and other molecular targets [44].

Conventional CS is generally considered to be ChT- and radiation-resistant, having limited treatment options when surgery is not feasible. As previously mentioned, recent studies reporting molecular genetic data have improved the understanding of CS biology, with some positive clinical findings already reported. Still, newer therapeutic targets are a critical unmet need.

## 6. Ewing Sarcoma

Ewing sarcoma (EWS), a high-grade mesenchymal neoplasm composed of sheets of small round blue cells with strong CD99 membrane expression, is characterized by a specific reciprocal chromosomal translocation merging a member of the FET family of proteins with several members of the ETS family of transcription factors (in 85–90% of cases, the chromosomal translocation t (11; 22) (q24; q12) is observed, generating the fusion product EWSR1-FLI1) [67,68,69]. EWS is an aggressive soft tissue and bone tumor accounting for 8% of all primary bone cancers. It mostly affects children (it represents 2% of all cancers in children), adolescents, and young adults (it is the second most common primary bone malignancy in this subgroup) and it has a peak incidence at the age of 15 years [67,68,69]. The incidence of EWS is higher in populations of European descent, and the tumor has male predominance (3:2–3:1 male:female ratio) [68,69]. EWS may develop in any body topography, with 80% of cases arising in bone tissue and 20% in extraosseous locations (more common in adults) [67,68,69]. 

Around 25% of all EWSs originate in the pelvis, with 20% of bone EWSs developing in the pelvic and sacral regions. A significant proportion of soft-tissue EWSs also develop in the pelvic and sacral regions, mostly from paravertebral soft tissues and gluteal muscles [67,68,69]. The pelvis is the second most common site of EWS, and primary pelvic EWS historically has the least favorable prognosis compared with all other sites, showing higher rates of local relapse and reduced survival rates (a consequence of the paucity of anatomic barriers to tumor diffusion and significant density of internal organs and neurovascular bundles verified in the pelvis, anatomical characteristics that make local control difficult) [14,70]. 

Several prognostic factors have been identified in EWS, which may be grouped as clinical and biological [71]. Clinical prognostic factors include metastatic disease at diagnosis as the most relevant, but also tumor volume, degree of tumor necrosis after neoadjuvant ChT, skeletal location, axial location, progression while receiving initial therapy, short remission, and patient age [71]. Biological prognostic factors include translocation type, number of copy number variations (CNV), CNV in 9p21, TP53 mutations, stromal antigen 2 mutations, CCL21 expression levels, and PD-1/PD-L1 lymphocytes/EWS expression [71]. 

Although the analysis of specific prognostic factors in pelvic EWS is scarce, it is of utmost importance to improve patient stratification to treatment, avoiding over- or under-treatment and allowing adoption of earlier risk- and response-adapted strategies. Chen and colleagues identified age, race, tumor stage, and surgery as independent prognostic factors of OS in pelvic EWS patients, while sex, tumor size, and RT were not significant predictors [70]. Younger age, Caucasian ethnicity, localized tumor stage, and undergoing surgery were associated with better prognosis [70]. In addition, the authors developed a nomogram (C-index of 0.728) capable of individually predicting 3 and 5 year OS in patients with pelvic EWS [70]. 

The current treatment of pelvic EWS relies on a systemic approach with induction/neoadjuvant ChT followed by local control strategies, including neoadjuvant or adjuvant RT and surgery. In selected cases, adjuvant/consolidation ChT may also be proposed. The choice of specific systemic regimens and local control strategies and the temporal sequence of each therapeutic approach are partially based on EWS risk groups, defined by broad EWS prognostic factors that serve as stratification criteria [69]. Standard-risk patients include patients with localized disease, small tumors (<200 mL), and good histologic response (<10% of viable tumor cells); patients with small tumors in whom histologic response cannot be assessed; and patients with localized disease and histologic response to induction ChT [69]. High-risk localized patients include those with unfavorable histologic response, with >10% viable tumor cells, and with large tumors (>200 mL) in whom histologic response cannot be assessed [70]. Very high-risk metastatic patients comprise patients with disseminated disease [70]. 

Cases of primary unresectable pelvic EWS pose a particularly difficult challenge that should be methodically tackled (Table 5 and Table 6). 

**Table 5 cancers-14-02546-t005:** Summary of therapeutic options for the management of unresectable pelvic Ewing sarcoma: evidence from clinical studies.

Therapeutic Modality	Therapeutic Relevance	Evidence Level	Comments
Chemotherapy [71]	++++	I	VDC/IE
Inhibition of IGF1/IGF1R loop ± temsirolimus [72,73,74,75]	++	IV	Ongoing research
Inhibition of IGF1R + Erlotinib [76]	++	IV	Ongoing research
Inhibition of IGF1R + Imatinib [77]	++	IV	Ongoing research
Regorafenib ± vincristine and irinotecan [71]	++	IV	Ongoing research
Cabozatinib [34]	++	IV	Ongoing research
Ganitumab + VDC/IE [78].	+	IV	Addition of ganitumab to VDC/IE did not improve survival
Radiotherapy [14,79,80,81]	++++	I	Neoadjuvant, adjuvant, and definitive setting

+ Not relevant; ++ scarcely relevant; ++++ highly relevant. IGF1, insulin-like growth factor-1; IGF1R, insulin-like growth factor-1 receptor; VDC/IE, vincristine doxorubicin cyclophosphamide/ifosfamide etoposide.

**Table 6 cancers-14-02546-t006:** Summary of therapeutic options for the management of unresectable pelvic Ewing sarcoma: evidence from pre-clinical studies.

Therapeutic Modality	Therapeutic Relevance	Evidence Level	Comments
Inhibition of EWSR1/FLI1 fusion protein (YK-4-279) [82]	+	IV	Resistance observed in murine models
Tazemetostat ± irinotecan or etoposide [83])	++	IV	Ongoing research
Inhibition of BET proteins + PI3K/mTOR inhibitor (BEZ235) [84]	++	IV	Ongoing research
Inhibition of LSD1 (HCI2509) [85]	++	IV	Ongoing research
Vorinostat + Temozolomide + Irinotecan [86]	++	IV	Ongoing research
Inhibition of CDK4/6 [87]	++	IV	Ongoing research
Inhibition of protein kinase C beta [71]	++	IV	Ongoing research
Inhibition of HSP90 + bortezomib [88]	++	IV	Ongoing research
Methylseleninic acid [89]	++	IV	Ongoing research
Trabectedin + IGF1 inhibitors [90]	++	IV	Ongoing research
Lurbinectedin + irinotecan [91]	++	IV	Ongoing research
Mithramycin analogues (EC-8105/EC-8042) [92]	++	IV	Ongoing research
Midostaurin + IGF1R inhibitors [93]	++	IV	Ongoing research
PARP inhibitors [71] ± trabectedin [94] ± radiotherapy [95]	++	IV	Ongoing research
Imatinib + doxorubicin [96]	++	IV	Ongoing research
Imatinib + cisplatin [97]	++	IV	Ongoing research
Regorafenib ± vincristine and irinotecan [71]	++	IV	Ongoing research
Cabozatinib [34]	++	IV	Ongoing research
Immunotherapy [14]	+	IV	Few, if any, responses seen with ICI. CAR-T cells are under evaluation
All-trans retinoic acid + EZH2 inhibitors/antibodies targeting HGF/agents targeting ganglioside GD2 [98,99,100]	+	IV	Ongoing research

+ Not relevant; ++ scarcely relevant. CAR, chimeric antigen receptor; CDK4/6, cyclin-dependent kinase 4/6; EZH2, enhancer of zeste homolog 2; HGF, hepatocyte growth factor; HSP90, heat shock 90 kDa protein; ICI, immune checkpoint inhibitor; IGF1, insulin-like growth factor-1; IGF1R, insulin-like growth factor-1 receptor; LSD1, lysine-specific demethylase-1; PARP; poly (ADP-ribose) polymerase; PI3K/mTOR, phosphatidylinositol 3-kinase/mammalian target of rapamycin; VDC/IE, vincristine doxorubicin cyclophosphamide/ifosfamide etoposide.

Neoadjuvant ChT is employed with the purpose of reducing the primary tumor size and targeting micrometastatic disease [67]. Neoadjuvant ChT is an integral component of the management of unresectable pelvic EWS, aiming not only to eliminate micrometastatic disease but also to promote cytoreduction, with the goal of turning primarily unresectable into resectable tumors. 

The isolated use of local control strategies, namely RT and/or surgery, is associated with very high rates of primary and/or distant relapse [14]. 

The golden rule of applying a combination of cytotoxic drugs to more effectively target EWS cells is evident in the standard neoadjuvant ChT backbone used in North America of interval-compressed VDC/IE (alternating VDC/IE cycles in 2 week intervals for 14 cycles) [14]. In Europe, the EURO Ewing 2012 trial compared vincristine, ifosfamide, doxorubicin, and etoposide (VIDE) with interval-compressed VDC/IE induction, with the latter showing a high probability of superiority according to preliminary data results [14]. 

The gradual optimization of current ChT regimens (neoadjuvant interval-compressed VCD/IE is an exquisite example) gradually pushed regimens towards the maximum tolerable intensity but still failed to cure an important proportion of patients and left survivors with a non-negligible load of late side effects [14]. The development of EWS-targeted therapies exploring vulnerabilities based on acknowledged mechanisms seems to be the most reasonable approach to improve outcomes and reduce the burden of dose-intense, side effect-prone ChT regimens [14]. The search for targetable biological markers has become the cornerstone of EWS research.

EWSR1/FLI1 is theoretically a promising treatment target. Direct inhibition of the fusion protein is hampered by its lack of enzymatic activity and disordered structure [67]. The small molecule YK-4-279 is an example of direct targeting of this fusion protein, having shown the ability to disrupt EWRS1/FLI1 and RNA helicase A protein binding, stopping EWSR1/FLI1 transcriptional activity in vitro by blocking EWSR1/FLI1 interactions with the spliceosome [82]. Unfortunately, drug resistance was observed in some murine models [101], and no clinical trials are currently being pursued in this setting. 

Therefore, effective therapeutics will have to rely on alternative mechanism-based approaches, targeting EWSR1/FLI1 downstream molecules, effector molecules of the EWSR1/FLI1 fusion protein, molecules and signaling pathways supporting and cooperating with the EWSR1/FLI1 fusion protein, or a combination of these approaches [67,71]. 

Examples of efforts to target downstream EWSR1/FLI1 and EWSR1/FLI1 effector molecules include enhancer of zeste homolog 2 (EZH2) inhibition (namely with tazemetostat, a highly specific EZH2 inhibitor, in combination with irinotecan or etoposide [83]), BET protein inhibition (namely with specific inhibitors of these proteins in combination with the PI3K/mTOR inhibitor BEZ235 [84]), lysine-specific demethylase 1 (LSD-1) inhibition (namely with the LSD-1 inhibitor HCI2509 [85]), NKX2.2 downregulation (namely with the histone deacetylase (HDAC) inhibitor vorinostat in combination with temozolomide and irinotecan [86]), cyclin D4 (CDK4) and D6 (CDK6) gene inhibition (namely with CDK 4/6 inhibitors [87] in monotherapy or combination with other synergistic agents), protein kinase C beta inhibition [71], heat shock 90 kDa protein (HSP90) inhibition (namely with HSP90 inhibitors in combination with bortezomib [88]), and Forkhead box O1 (FOXO1) induction (namely with methylseleninic acid [89]) [71]. 

A non-specific downstream targeting approach is also plausible [71], namely through the combination of trabectedin and insulin-like growth factor-1 (IGF1) inhibitors [90] or second-generation lurbinectedin with irinotecan [91], the use of mithramycin analogues (EC-8105 and EC-8042) [92], or the combination of midostaurin with insulin-like growth factor-1 receptor (IGF1R) inhibitors [93]. 

Additional strategies include modulating signaling pathways that support and cooperate with the EWSR1/FLI1 fusion protein. Inhibition of IGF1/IGF1R loop with monoclonal antibodies (such as R1507 [72], cixutumumab [73], and figitumumab [74]) in monotherapy or in combination with mTOR inhibitors (namely temsirolimus [75]), HSP90 inhibitors [88], erlotinib [76], imatinib [77], and CDK4/6 inhibitors [102] may be promising alternatives. Inhibition of poly-ADP-ribose (PARP) with PARP inhibitors (such as olaparib, talazoparib, and niraparib) in monotherapy [71] or in combination with trabectedin [94] or RT [95] has also shown encouraging results [71]. Tyrosine kinase receptors (c- KIT and PDGFR) inhibitors, namely imatinib in combination with doxorubicin [96] or cisplatin [97], regorafenib in monotherapy or in combination with vincristine and irinotecan [71], and cabozantinib [34] may also represent a valid approach. The use of ataxia telangiectasia and Rad-3-related protein (ATR) inhibitors and VEGFR inhibitors is also being currently assessed [71]. 

Unfortunately, the great majority of the above-mentioned biological target inhibitors or inductors have only been tested in preclinical models. Only a very small number of biological modulators have been or are currently being tested in early-phase clinical trials, mostly involving patients with refractory/relapsed or metastatic EWS. The number of targeted agents that are currently being assessed for induction/neoadjuvant use in EWS is particularly scarce. 

It is crucial to add supplementary cytoreductive power to backbone regimens. The focus must be placed on the development of additional extra-targeted therapies that can be combined with neoadjuvant regimens to maximize the tumor volume shrinkage effect. Targeting the IGF1-axis with this intent is a previously tested strategy, with the use of IGF1R inhibitors leading to consistent response rates of about 10% [72,103,104]. Within the same strategy, the IGF1R monoclonal antibody inhibitor ganitumab was combined with interval-compressed VDC/IE in the randomized phase III AEWS1221 trial in patients with metastatic classical EWS, with no improvement shown in survival rates with the addition of ganitumab to ChT [78].

The use of immunotherapy in EWS is limited, irrespective of the disease setting. EWS per se carries a low mutational burden and has one of the lowest mutational rates of all tumors (0.15 mutations per megabase) [67]. The use of immunotherapy in EWS must overcome additional challenges, such as the lack of HLA class I expression and the immunosuppressive tumor microenvironment due to the presence of myeloid-derived suppressor cells, F2 fibrocytes, and M2-like macrophages [14]. The use of ICIs, such as anti-PD-1/PD-L1 or anti-CTLA4 antibodies, led to few, if any, responses, even though a significant number of trials are currently taking place [14]. Several cell-based immunotherapy strategies (adoptive T-cell transfer and chimeric antigen receptor (CAR) T-cells) are also being explored in clinical trials [14]. Putative targets for T-cell-based strategies include cell-surface molecules, such as EGFR/HER2, IGF1R, and ROR1 [14]. Combinations of all-trans retinoic acid, EZH2 inhibitors, and antibodies targeting the hepatocyte growth factor (HGF) with agents targeting the ganglioside GD2 (an already established immunotherapy target in neuroblastoma) have shown promising results in preclinical studies [98,99,100]. 

EWS radiosensitivity has been acknowledged since the first report by James Ewing [105]. RT may be an option for local control within a definitive treatment modality, mainly in the case of unresectable tumors, but may also be delivered in combination with surgery, either pre- or postoperatively [14]. Moreover, it has well-acknowledged utility when used with palliative intent [14]. In general terms, preoperative or neoadjuvant RT is the preferred option in cases of tumor progression after neoadjuvant systemic treatment or anticipated marginal or intralesional resection. The AEWS1031 Ewing protocol recommends preoperative RT for apparently resectable tumors in selected sites, such as the pelvis and the chest wall, and axial tumors with a high risk of positive resection margins (R+ resection) [14]. Postoperative (or adjuvant) RT is advocated for cases of intralesional (comprising intraoperative spill) or marginal (R1 resection) surgery. European protocols also favor postoperative RT in cases of poor histologic response (>10% of viable tumor cells within the resected tissue), regardless of surgical margins [14]. Definitive RT is conceptually destined for tumors labeled as unresectable or smaller tumors (<8 cm) [79,80]. 

The role of RT in localized EWS management has evolved. Most of the evidence from the last 20–30 years pointed towards the superiority of surgery compared to isolated RT regarding both local control and OS [106], while recent data support the notion that a combined treatment strategy with surgery and adjuvant RT seems to offer synergistic effects and improved local control for EWS patients without wide surgical margins or with poor histologic response to neoadjuvant ChT [106]. As previously mentioned, pelvic EWS carries the least favorable prognosis of all EWS topographies [14]. However, recent studies showed increased local control and survival rates with combined surgery and RT [106,107]. Pelvic EWS patients presenting with large pelvic tumors (>200 mL) seem to benefit most from combined local modalities [108,109]. 

RT timing and technique are of paramount importance in the management of pelvic EWS (primarily unresectable or resectable). Based on EURO Ewing 2012 and COG AEWS1031 guidelines, definitive RT should be offered to pelvic EWS patients with unresectable tumors [14]. Preoperative RT is recommended for apparently resectable pelvic tumors with high R+ risk (preferably based on the post-induction ChT MRI) and optimizes the induction ChT cytoreductive effect, specifically of the extraosseous component, increasing the possibility of limb-sparing surgery [14]. Postoperative RT is an option not only in cases of intralesional (including intraoperative spill) or marginal (R1 resection) surgery, but also in cases of R0 resection, in cases of wide resection (based on histology) and viable tumor cells > 10%, marginal resection (based on surgery) and viable tumor cells < 10%, or tumor resection prior to ChT without subsequent excision of all initially involved tumor tissue [14]. 

The relevance of preoperative (or neoadjuvant) RT in the optimal management of pelvic EWS has progressively increased. Shuck et al. and Donati et al. reported that pelvic EWS patients receiving preoperative RT displayed increased local control rates compared to those undergoing surgery with or without postoperative RT [110,111]. Recently, Lex et al. showed that non-selective preoperative photon- or proton-beam RT has the potential to improve local recurrence-free survival compared to selective delivery of postoperative RT in pelvic EWS [112]. In this study, patients receiving preoperative RT had the highest proportion of wide margins [112]. Preoperative RT may reduce the risk of positive microscopic margins by increasing rates of cell necrosis and enabling technically easier surgical resections by providing a clearer zone of pathologic tissue, with higher probability of achieving functional, limb-saving surgery [112]. The choice of the optimal RT technique may further reduce toxicity. 

Risks associated with photonic RT include delayed wound healing, infection, fibrosis, fracture of the irradiated bone, osteoarthritis, and secondary RT-induced malignancy (radiation-induced sarcomas and hematologic malignancies) [112]. Children are particularly vulnerable to radiation-induced late toxicities and secondary malignancies due to the immature nature of their tissues [112].

Reduction of treatment areas and dose burden to healthy tissues is the gold principle of modern, very high-precision RT. 

Intensity-modulated RT (IMRT) and proton beam therapy (PBT) are modern techniques that provide optimal tumor coverage and sparing of critical structures [14,112]. In pelvic EWS, the evidence shows greater dose conformity after modern IMRT than after conventional photon-based RT [14]. Results of PBT in EWS treatment are auspicious [14]. Early experiments suggest even higher conformity with intensity-modulated PBT (IMPBT) compared to IMRT, namely when considering complex target volumes of pelvic EWS lesions. These may significantly benefit from PBT, due to the steep dose fall-off distally to the target and relatively low number of treatment beams required for optimal dose conformity [14]. PBT became a standard tool in modern RT, mostly in young patients, to reduce the risk of late RT effects and in cases with curative intent [14]. Recently, Uezono et al. reported that photon therapy offers local control compared to photon therapy in pediatric patients with pelvic EWS [81]. 

Strengthening the cytoreductive power of neoadjuvant therapeutic modalities with tolerable and acceptable toxicity is a key goal in the efforts to improve resectability of primarily unresectable pelvic EWS. The backbone of neoadjuvant treatment may soon include new targeted therapies and T-cell-based immunotherapies. A multimodal induction strategy comprising the cytoreductive potency of conventional optimized ChT regimens, exponentiated by its combination with specific targeted therapies, possibly enriched by the concomitant or sequential immunobiological modulation of T-cell-based immunotherapies, and synergically catalyzed by sequential dose conformity-enhanced modern RT techniques (such as PBT) will probably change recurrent disease prognosis and survival rates in pelvic EWS. New targeted therapies and state-of-the-art T-cell-based immunotherapies may also be used within a definitive treatment strategy for pelvic EWS patients, shaping the treatment landscape of the disease. 

## 7. Chordoma

Chordomas are low-grade notochordal tumors that frequently arise from the axial skeleton and have a peak incidence at 50–60 years of age [113]. Despite being the most common primary sacral tumors (>50% of total tumors), only 29.2–50% of all chordomas have sacrococcygeal presentation, with male predominance (~2:1) [113,114]. Histologically, chordomas are characterized by a lobulated architecture composed of epithelioid cells arranged in cords and nests within a myxoid matrix. Cells have vacuolized cytoplasm, which may present as single to multiple vacuoles, creating a bubbly appearance (physaliferous cells). A component of high-grade spindle and/or pleomorphic sarcoma is rarely present and establishes the diagnosis of dedifferentiated chordoma. Tumor cells are typically positive for keratin, EMA, branchury, and occasionally S100 [115]. The differential diagnosis includes metastatic carcinoma, chondrosarcoma, chordoid meningioma, and myoepithelial tumor of bone. 

Symptoms tend to appear late in the course of disease due to these tumors’ indolent growth. When patients do become symptomatic, they generally complain of localized (visceral) pain, radiculopathy, myelopathy, and/or bowel/bladder dysfunction. Although surgery is recommended whenever possible, wide or marginal resection may not be feasible. The extent of excision generally goes beyond sacrectomy and carries an associated risk of incontinence, sexual dysfunction, saddle anesthesia, impaired mechanical stability, wound dehiscence, wound infection, and cerebrospinal fluid leak [113,116]. The rate of wide or marginal margin resection in sacral chordoma ranges from 40–55.6% and has a negative impact on patients’ quality of life [116,117]. OS and cancer-specific survival should be considered, but published nomograms are scarce [118]. The nomogram proposed by Zheng considers size (>29 mm), age (>55 years), and histology (dedifferentiated chordoma) as worse prognostic factors for survival in patients with pelvic presentation [114].

Treatment options for unresectable cases vary according to whether the patient has been previously submitted to local treatment (Table 7). In *naïve* patients, local treatment solely with RT can be an option, but because most data have not been retrieved from randomized controlled trials, result interpretation is not free of bias, and the debate regarding the preferred treatment approach is ongoing. Most data suggest that an irradiation dose > 70 Gy (relative biological effectiveness; RBE) confers increased local control and OS regardless of the RT modality [116,119,120,121]. Hadrontherapy with proton or carbon ions seems to result in higher local control rates but not significantly different OS rates compared to conventional IMRT [116]. The use of moderately hypofractionated schedules for high linear energy transfer radiation (e.g., carbon ions) can be an option. Ion carbon hypofractionation prescription differs between Japanese and European centers due to different RBE models [119]. It should be kept in mind that favorable RT results are achieved when tumor volume is low. This is a general rule of thumb regardless of tumor histology, demonstrated in a large retrospective chordoma study [117] that spurred efforts to use RT as adjuvant or, more recently, neoadjuvant treatment. The 2017 Milan Chordoma Global Consensus Group’ defined the appropriate surgical and RT margins for sacral chordoma [119]. The ongoing SACRO trial (NCT02986516) is recruiting patients to assess relapse-free survival in primary localized sacral chordoma treated with surgery versus definitive RT. This is a multicenter, comparative, open-label, parallel-group, mixed observational-randomized controlled trial that randomizes patients to receive treatment A (surgery, with or without RT) or treatment B (definitive RT). Patients who refuse randomization will be included in the Prospective Cohort Study and treated according to their choice (treatment A or B). This study will be important to define whether there is equipoise between these treatments.

Patients with local recurrence who are deemed unsuitable for treatment with curative intent (RT or surgery) should be offered best palliative care, according to the Chordoma Global Consensus Group [120]. Palliative local treatment options comprise debulking surgery, low-dose RT, radiofrequency ablation, and other locoregional approaches (i.e., cryotherapy) to relieve local symptoms. Radiofrequency ablation (RFA) is considered to be a safe minimally invasive technique, which offers potential local control and pain relief for bone tumors [118,125]. This technique, based on frictional heating, has the ability to effectively destroy tumor tissue; however, extreme care is paramount whenever facing larger or previously irradiated tumors, which increases the possibility of further complications [118,126,127]. Cryotherapy is based on the principle of cooling tumor tissue to −20 °C, which induces cell death in the tissues immediately adjacent to the inserted probe through intracellular ice formation [125,128,129]. Also, at a further distance, a gradual cooling occurs, which is the cause of osmotic differences across the cell membrane, with secondary cellular dehydration and subsequent cell death [128]. Additionally, and despite the fact that this technique is considered effective against primary and secondary bone tumors, there is a significant complications rate, namely due to peripheral bone necrosis and cold injury in the surrounding soft tissues [129,130]. Before starting systemic therapy, the Consensus Group recommends a brief observation period to properly document progression, as treatment options are still limited [120]. 

Given the chemoresistant histologic nature of these tumors, systemic treatment for chordoma patients in this setting mostly relies on targeted therapy [120]. Current clinical research focuses on targeting platelet-derived growth factor receptors (PDGFR), stem cell factor receptor (KIT), epidermal growth factor receptor (EGFR), erbB-2/human epidermal growth factor receptor 2 (HER2/neu), VEGFR, and phosphatidylinositol 3-kinase/protein kinase B/mammalian target of rapamycin (PI3K/AKT/mTOR) pathway [113,122]. In the current practice, imatinib and sorafenib are the most studied agents for the treatment of advanced chordoma and a reasonable option for slow-growing tumors [120]. A recent phase II trial assessed the antitumor activity of 800 mg/day of imatinib until progression in 56 patients with advanced PDGFβ and/or PDGFRβ-mutated chordoma [123]. The median PFS was 9 months, and one patient had partial response at 6 months (overall response rate, 2%). In addition, 35 patients (70%) had stable disease. No unexpected toxicities were observed [123]. Another phase II trial assessed the activity of 800 mg/day of sorafenib in 27 patients [124]. The best objective response was observed in one patient (3.7%), and the study achieved a 9 month PFS rate of 73.0% and a 12 month OS rate of 86.5% [124]. Research is also ongoing on ICIs, brachyury (cancer vaccines targeting drivers of epithelial–mesenchymal transition), and poly (ADP-ribose) polymerase (PARP) inhibitors [113,122,131,132].

Access to innovative treatments for this tumor type is disparate between countries, and basic/clinical research is scattered worldwide. The Chordoma Foundation is a major pillar in mitigating these disparities, aggregating data about ongoing trials for patients/caregivers and health professionals and promoting research (innovative drug and drug repurposing initiatives) and industry investment.

## 8. Discussion

The pelvis remains an elusive anatomic site for effective targeted therapeutic approaches in PBS. The pelvic topography is characterized by a shortage of anatomic barriers to BS extension and infiltration, affecting a remarkable number of tightly packed crucial organs and an intricate system of noble vessels and pivotal nervous elements [70]. These structural constraints, together with the biological nature of BSs most frequently arising in the pelvis (osteosarcoma, CS, EWS, chordoma), pose unique challenges for the multidisciplinary sarcoma team: surgeons struggle to perform negative -margin resections and reconstruct functional limbs; radiation oncologists seek to balance the radiation dose capable of achieving appropriate local control with the collateral damage to neighboring vital organs using emerging IMRT and IMPB therapies; and medical oncologists seek new targets for immunomodulatory and targeted therapies, as the intensification limits of conventional ChT have been largely met. 

Irrespective of the specific sarcoma type, PBS is typically characterized by late symptomatic presentation, metastatic expression upon diagnosis, challenging surgical approaches due to intricate pelvic anatomy, and high degrees of chemo- and radioresistance, leading to poor outcomes and dismal prognosis. 

If resection is feasible after appropriate cytoreduction, primarily unresectable PBS is tackled with a neoadjuvant approach, using either combination ChT regimens (such as MAP in osteosarcoma; VDC/IE, doxorubicin, and cisplatin in CS (especially in mesenchymal and dedifferentiated CS); or VDC/IE in EWS), conventional photonic RT or carbon or proton hadrontherapy in specific osteosarcoma or EWS cases where progression is documented after neoadjuvant ChT or marginal or intralesional resection is predicted. Isolated immunotherapy modalities (ICIs or CAR-T cells) and isolated or neoadjuvant ChT-combined targeted approaches are currently being assessed in this setting.

RT plays a crucial adjuvant role in consolidating the outcomes of surgical resection, both in osteosarcoma (after incomplete resection) and EWS. In EWS, RT is an option not only in cases of intralesional or marginal surgery (R1 resection) but also in cases of R0 resection upon wide resection (based on histology) and viable tumor cells > 10%, marginal resection (based on surgery) and viable tumor cells < 10%, or tumor resection prior to ChT without subsequent excision of all initially involved tumor tissue [14]. 

Definitive and palliative strategies are of paramount importance in cases of absolute unresectability defined by size or location (with relevant surgical morbidity or difficulty/impossibility in achieving clear margins) and of additional chemoresistance (e.g., in chondroblastic osteosarcoma, low-grade CS, chordoma). In cases of chemoresistance, photonic RT (IMRT or IMPB) or hadrontherapy (with proton or carbon ions) are good options, as already shown in osteosarcoma, mesenchymal CS, EWS, and chordoma. Radioresistance, observed, for example, in low-grade CS, is an additional issue. The use of TKIs, mTOR inhibitors, CDK 4/6 inhibitors, and other targeted therapies, as well as immunotherapy (ICIs and CAR-T cells), has shown promising results in different PBS types in this setting. Lastly, palliative approaches, including the use of radionucleotides (e.g., 153-Sm-EDTMP in osteosarcoma), embolization, radiofrequency ablation and cryotherapy (e.g., in chordoma), are of great relevance in the management of these tumors. 

The future treatment landscape of primarily unresectable PBS will be shaped by the development of targeted therapies (inhibitors and monoclonal antibodies) and immunotherapies (ICIs and cell-based immunotherapy) for specific sarcoma subtypes, aiming to improve the cytoreductive power of conventional ChT with acceptable toxicity and ameliorate local and symptomatic control in definitively unresectable cases. New intensity-modulated conventional photonic RT and hadrontherapy (proton beam and ion carbon RT) provide better conformation, with optimal tumor coverage and sparing of critical structures.

## 9. Final Remarks

The management of unresectable PBS remains a challenge. Despite the recent emergence of various targeted therapies and progress in RT modalities, it is crucial to provide additional cytoreductive power to conventional regimens and improve neoplastic and symptomatic control in definitive unresectable disease. The development of specific targeted and immune therapies, together with the optimization of photonic RT and hadrontherapy, will desirably pave the way to improved patient outcomes. 

## Figures and Tables

**Figure 1 cancers-14-02546-f001:**
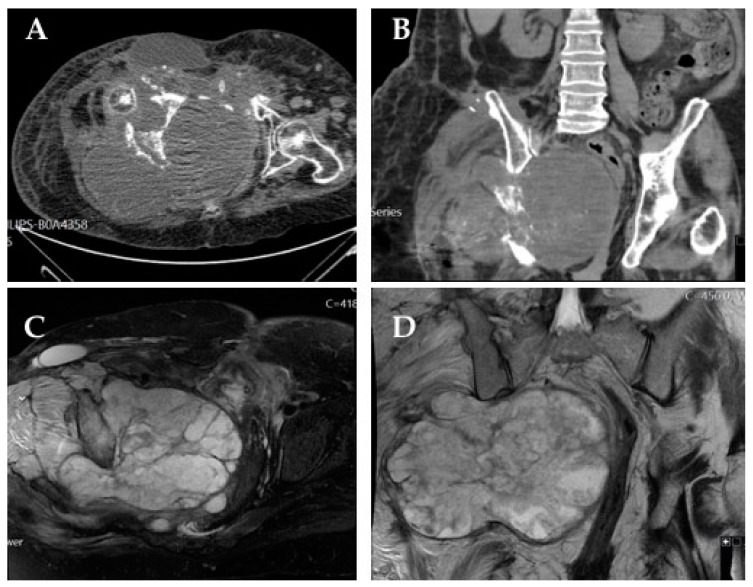
Very large chondrosarcoma arising from the right hemipelvis: axial (**A**) and coronal (**B**) computed tomography (CT) scan images; fat-suppressed T2-weighted axial (**C**) and coronal (**D**) magnetic resonance imaging (MRI). The patient was managed with definitive radiotherapy, with overall survival of 3 years.

**Figure 2 cancers-14-02546-f002:**
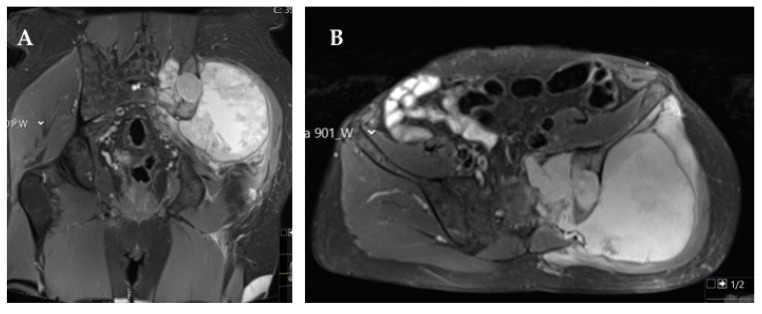
Magnetic resonance imaging (MRI) fat-suppressed T2-weighted coronal (**A**) and axial (**B**) images showing a large chondrosarcoma occupying the left hemipelvis infiltrating the left sacro-iliac joint and sacrum until the midline. This patient was managed with definitive radiotherapy due to refusal to accept the potential functional impairment and complications associated with the surgical procedure.

**Figure 3 cancers-14-02546-f003:**
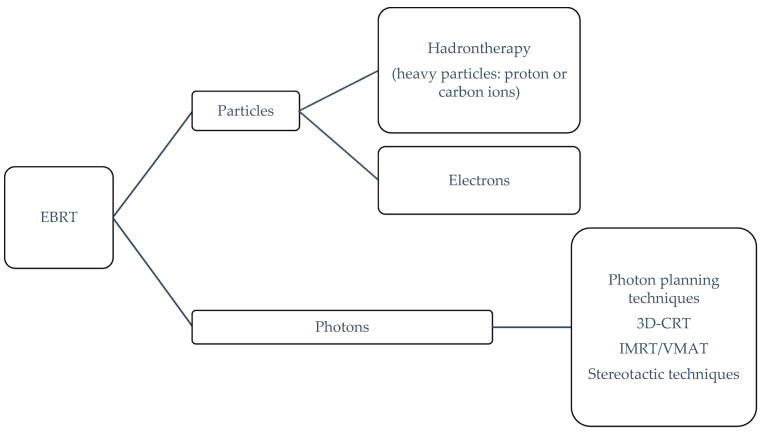
EBRT classification.

**Table 1 cancers-14-02546-t001:** Radiotherapy terms adapted from Ref. [16].

Term	Definition
3D-CRT	A human operator (i.e., a dosimetrist) generates the best beam arrangement to encompass the target. This approach is known as forward planning
IMRT	Each beam arrangement has a non-uniform fluence/intensity and the approach is based on inversed planning: the operator feeds the planning system the desired dose for the target and OAR restrictions upfront. The software performs multiple iteration to search for the best and optimized solution for beam arrangement
VMAT	Like IMRT, but beams with varying intensities are delivered when the gantry is rotating around the patient
LET	Total amount of energy deposited per unit distance in biological materials by ionizing radiation
RBE	Radiation relative biological effectiveness depending on LET, type of radiation particle, total dose, and dose fractionation
Proton active scanning	Proton (charged particle) beam delivery based on a pencil beam that is steered using a magnet in the beam line. The dose is then deposited layer by layer
Intensity-modulated proton therapy	Further shapes the active scanning to the distal tumor for irregularly shaped tumors

**Table 7 cancers-14-02546-t007:** Summary of therapeutic options for the management of unresectable pelvic chordoma: evidence from clinical studies.

Therapeutic Modality	Therapeutic Relevance	Evidence Level	Comments
Photon radiotherapy [116,117]	+++	III/III	Best outcomes with small tumor target volumes
Proton and carbon ion radiotherapy [115,116]	++++	IV/III	Best outcomes with small tumor target volumes
Chemotherapy [120]	+		
Imatinib [122,123]	+++	III/II	In PDGFβ- or PDGFRβ-positive chordoma
Sorafenib [124]	+++	III/II	

+ Not relevant; +++ relevant; ++++ highly relevant. PARP, poly (ADP-ribose) polymerase; PDGFβ, platelet-derived growth factor subunit β; PDGFRβ, platelet-derived growth factor receptor subunit β.

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
