# Peer review of "Management of Unresectable Localized Pelvic Bone Sarcomas: Current Practice and Future Perspectives"

_cancers, 2022, doi:10.3390/cancers14102546_

Round 1

Reviewer 1 Report

The manuscript is well-structured, clear, and cohesive. The studies cited are relevant to the topic, comprehensive, unbiased, and up-to-date.  Authors did a good job summarizing most of the previous and current pre-clinical and clinical studies relevant to bone sarcomas.

Revision recommendations:

  1. Each section includes one to two paragraphs of very basic information (e.g. epidemiology, pathology, clinical presentation), which may not be necessary given the targeted reader population. This information could be more succinct and concise.
  2. This review includes a mixture of pre-clinical and clinical studies. While the preclinical data do provide insights into future perspectives of practice, I find it slightly confusing when some in vitro and in vivo studies are summarized along with clinical trials in the same table. Consider discussing preclinical studies in one separate paragraph/table under each tumor type.
  3. An explanation of how "therapeutic relevance" (in each table) was systematically scored is needed.

Author Response

We much appreciate the opportunity to be able to address the reviewers’ comments and resubmit our manuscript. Furthermore, we were delighted with the reviewers ‘thorough reading, helpful suggestions and both commending the importance of the work presented.

Below for reviewer 1 we have addressed the concerns and suggestions as well as possible and for convenience have referred to the manuscript lines where improvements have been made.

Reviewer 1

“The manuscript is well-structured, clear, and cohesive. The studies cited are relevant to the topic, comprehensive, unbiased, and up-to-date.  Authors did a good job summarizing most of the previous and current pre-clinical and clinical studies relevant to bone sarcomas”.

RESPONSE: We thank this reviewer for the positive message and the helpful suggestions to improve our manuscript.

“Revision recommendations:

  1. Each section includes one to two paragraphs of very basic information (e.g. epidemiology, pathology, clinical presentation), which may not be necessary given the targeted reader population. This information could be more succinct and concise”.

RESPONSE: We thank the reviewer for this critical, but very correct assessment. Regarding this specific recommendation, we opted to shorten the extension of the first paragraph of the section relative to osteosarcoma (page 5, lines 159-168) and also the extension of the second paragraph of the section relative to Ewing sarcoma (page 11, lines 419-427).

  1. “This review includes a mixture of pre-clinical and clinical studies. While the preclinical data do provide insights into future perspectives of practice, I find it slightly confusing when some in vitro and in vivo studies are summarized along with clinical trials in the same table. Consider discussing preclinical studies in one separate paragraph/table under each tumor type”.

RESPONSE: We thank the reviewer for this helpful suggestion, we should have distinguished the nature of the cited studies and trials more clearly. This was more significantly verified in the Ewing sarcoma section, therefore, in line with the reviewer suggestion, we opted to present two separate tables: the first one regarding the evidence from clinical studies and the second one regarding the evidence from pre-clinical studies as it may be seen in pages 15-16, lines 650-684. Besides this, we also opted to assume a similar strategy for the tables that summarize the content of the chondrosarcoma section as it may be seen in page 10, lines 387-397. All the data/ information relative to therapeutic options listed in the osteosarcoma and chordoma sections were obtained from clinical studies, therefore we opted to highlight this fact in the respective tables’ titles (osteosarcoma table – page 7, line 250; chordoma table – page 18, line 789).

  1. “An explanation of how "therapeutic relevance" (in each table) was systematically scored is needed”.

RESPONSE: We, once again, thank the reviewer for this precious assessment. We agree with the imperative necessity to provide a thorough explanation of the criteria we used to systematically score the therapeutic relevance of each intervention. Effectively, an explanation is specifically comprehensively provided in the “Introduction” section (page 2, lines 57-76).

Reviewer 2 Report

Summary:

This manuscript is a review of the management of unresectable localized pelvic bone sarcomas. The authors delineate the challenges of treating primary bone sarcomas in the pelvis and sacrum due to anatomic constraints and difficulty in obtaining negative margins. They review each major bone sarcoma histology affecting the pelvis and sacrum in detail as well as the systemic treatment and radiation options in detail for each of these in the setting of unresectable disease. The review is thorough, and the management of “unresectable” disease would add to this special issue on Pelvic and Sacral Bone Sarcoma Diagnosis and Treatment in Cancers.

Major Issues:

While giant cell tumor of bone can be locally aggressive, difficult to treat (esp. in sacral locations), and in rare cases “embolize” to the lung it should not be included as a bone sarcoma with this group. Personally, would restrict this review to Conventional Osteosarcoma, Ewing sarcoma, Chondrosarcoma, and Chordoma.  

As radiation is a common modality in the treatment of unresectable disease across all the histologies reviewed in this manuscript, a separate section should be used to review radiation modalities from IMRT to heavy particle (proton and carbon ion, etc). This should be in a section after 2. Definition of unresectable pelvic sarcoma.

While there is not a large body of evidence for palliative thermal ablation options like RFA and cryotherapy, there may be value in a paragraph presenting these modalities rather than first mention toward the end of the discussion.

Would add some more figures as example of other cases highlighting utilization of the treatments noted in this review for unresectable pelvic sarcomas. Thus far there is only the figure presenting a chondrosarcoma of the right hemipelvis that was deemed “unresectable” and treated with definitive RT.

In the therapeutic option tables, the authors may need to clarify the relevance column. How did they subjectively determine Not, Little, Relevant, and Highly relevant? Was it based on a certain PFS or response rate? They may want to highlight their methodology for relevance and level of evidence in the intro/methods of this review since it is presented in an evidence based manner.

Minor Issues:

While not of pelvic location, would cite COG paper on radiation for femur EWS : Daw et al Ann Surg Oncol 2016 for discussion of local control.

Awkward wording: line 64 – “PBSs often present notable volumetry and extension to neighboring organs, vessels … “

Grammar/spelling: Line 69 - :”if the patient does not accept the functional impairment …”

Need to insert malignant: Line 80: “… osteosarcoma is the most common primary malignant tumor of bone.”

Clarify: Line 95-96: “The presence of woven bone or osteoid in close association with surrounding cells is required for diagnosis.” – would reword a little to be clear that the malignant cells are producing the osteoid

Author Response

We much appreciate the opportunity to be able to address the reviewers’ comments and resubmit our manuscript. Furthermore, we were delighted with the reviewers ‘thorough reading, helpful suggestions and both commending the importance of the work presented.

Below and for reviewer 2, we have addressed the concerns and suggestions as well as possible and for convenience have referred to the manuscript lines where improvements have been made.

Reviewer 2

“This manuscript is a review of the management of unresectable localized pelvic bone sarcomas. The authors delineate the challenges of treating primary bone sarcomas in the pelvis and sacrum due to anatomic constraints and difficulty in obtaining negative margins. They review each major bone sarcoma histology affecting the pelvis and sacrum in detail as well as the systemic treatment and radiation options in detail for each of these in the setting of unresectable disease. The review is thorough, and the management of “unresectable” disease would add to this special issue on Pelvic and Sacral Bone Sarcoma Diagnosis and Treatment in Cancers.”

RESPONSE: We thank this reviewer for the kind assessment and the time set aside to improve our manuscript.

“Major Issues:

While giant cell tumor of bone can be locally aggressive, difficult to treat (esp. in sacral locations), and in rare cases “embolize” to the lung it should not be included as a bone sarcoma with this group. Personally, would restrict this review to Conventional Osteosarcoma, Ewing sarcoma, Chondrosarcoma, and Chordoma”.  

RESPONSE:  We understand and agree the reviewer suggestion and, therefore, the section relative to giant cell tumor of bone and the respective table were removed as it may be seen in page 18 (please note track changes). Besides this, naturally, all the references relative to giant cell tumor of bone and its therapeutic approach were removed from the “Discussion” section as it may also be seen in pages 18 and 19.

“As radiation is a common modality in the treatment of unresectable disease across all the histologies reviewed in this manuscript, a separate section should be used to review radiation modalities from IMRT to heavy particle (proton and carbon ion, etc). This should be in a section after 2. Definition of unresectable pelvic sarcoma”.

RESPONSE: We thank and salute the reviewer for this particularly helpful suggestion. We agree with the reviewer and a new section relative to radiation modalities amenable to be used in the treatment of pelvic bone sarcomas was introduced precisely after the section “Definition of unresectable pelvic sarcoma”, in line with the reviewer suggestion. The new section was entitled “Radiotherapy and its different modalities: A cornerstone of unresectable pelvic sarcomas management” and may be found in pages 4-5, lines 117-154.

“While there is not a large body of evidence for palliative thermal ablation options like RFA and cryotherapy, there may be value in a paragraph presenting these modalities rather than first mention toward the end of the discussion”.

RESPONSE: We agree with the reviewer and opted to include a paragraph summarizing the evidence relative to palliative thermal ablation options as it may be read in page 17, lines 740-752.

“Would add some more figures as example of other cases highlighting utilization of the treatments noted in this review for unresectable pelvic sarcomas. Thus far there is only the figure presenting a chondrosarcoma of the right hemipelvis that was deemed “unresectable” and treated with definitive RT”.

RESPONSE: We, once again, agree with the reviewer and added more exemplificative figures as it may be observed in page 3, lines 104-114.

“In the therapeutic option tables, the authors may need to clarify the relevance column. How did they subjectively determine Not, Little, Relevant, and Highly relevant? Was it based on a certain PFS or response rate? They may want to highlight their methodology for relevance and level of evidence in the intro/methods of this review since it is presented in an evidence based manner”.

RESPONSE: We thank the reviewer for addressing the necessity to clarify our criteria for the attribution of a therapeutic relevance level. As previously mentioned, an explanation is specifically comprehensively provided in the “Introduction” section (page 2, lines 57-76).

“Minor Issues:

While not of pelvic location, would cite COG paper on radiation for femur EWS : Daw et al Ann Surg Oncol 2016 for discussion of local control”.

RESPONSE: We agree with the reviewer and added a mention to the referred COG paper as it may be read on page 14, lines 630-633.

“Awkward wording: line 64 – “PBSs often present notable volumetry and extension to neighboring organs, vessels … “

RESPONSE: We again thank the reviewer for the work improving our manuscript. We’ve modified the formulation of the referred sentence as it may be read on page 2, line 85.

“Grammar/spelling: Line 69 - :”if the patient does not accept the functional impairment …”

RESPONSE: We thank the reviewer for spotting this typo and have corrected this as it is visible in page 2, line 90.

“Need to insert malignant: Line 80: “… osteosarcoma is the most common primary malignant tumor of bone.”

RESPONSE: We have inserted the word malignant as suggested and as it may be read in page 5, line 159.

“Clarify: Line 95-96: “The presence of woven bone or osteoid in close association with surrounding cells is required for diagnosis.” – would reword a little to be clear that the malignant cells are producing the osteoid”.

RESPONSE: We have reworded as suggested and as it may be read in page 5, lines 174-175.

We would also like to mention that, additionally, we adapted and homogenised the format of all references in line with what was requested. As recommended, we used the style files for Endnote provided in Cancers website.

Round 2

Reviewer 2 Report

The manuscript is improved and the authors have responded to reviewer comments and suggestions.

I would recommend deleting paragraph regarding EWS of proximal femur, page 14, lines 599-602. This paragraph doesn't add to the content and distracting since about proximal femur. The COG paper referenced can be deleted if doesn't fit or can be referenced in general RT effectiveness on primary location in EWS. 
